# Phase-referenced nonlinear spectroscopy of the α-quartz/water interface

Paul E. Ohno[1], Sarah A. Saslow[1,†], Hong-fei Wang[2], Franz M. Geiger[1] & Kenneth B. Eisenthal[3]

Probing the polarization of water molecules at charged interfaces by second harmonic generation spectroscopy has been heretofore limited to isotropic materials. Here we report non-resonant nonlinear optical measurements at the interface of anisotropic $z$-cut α-quartz and water under conditions of dynamically changing ionic strength and bulk solution pH. We find that the product of the third-order susceptibility and the interfacial potential, $\chi^{(3)} \times \Phi(0)$, is given by $(\chi_1^{(3)} - i\chi_2^{(3)}) \times \Phi(0)$, and that the interference between this product and the second-order susceptibility of bulk quartz depends on the rotation angle of α-quartz around the $z$ axis. Our experiments show that this newly identified term, $i\chi^{(3)} \times \Phi(0)$, which is out of phase from the surface terms, is of bulk origin. The possibility of internally phase referencing the interfacial response for the interfacial orientation analysis of species or materials in contact with α-quartz is discussed along with the implications for conditions of resonance enhancement.

[1] Department of Chemistry, Northwestern University, Technological Institute Room KG68, 2145 Sheridan Road, Evanston, Illinois 60208, USA. [2] Physical Sciences Division, Physical and Computational Sciences Directorate, Pacific Northwest National Laboratory, Richland, Washington 99352, USA. [3] Department of Chemistry, Columbia University, New York, New York 10027, USA. † Present address: Earth Systems Science Division, Energy and Environment Directorate, Pacific Northwest National Laboratory, Richland, Washington 99352, USA. Correspondence and requests for materials should be addressed to F.M.G. (email: geigerf@chem.northwestern.edu).

The strength of the second harmonic electric field, $E_{2\omega}$, that is produced at charged interfaces is a function of the incident fundamental electric field, $E_\omega$, the second-order susceptibility of the interface, $\chi^{(2)}$, the zero-frequency electric field corresponding to the interfacial potential produced by surface charges, $\Phi(0)$, and the third-order susceptibility, $\chi^{(3)}$, according to[1-6]

$$\sqrt{I_{SHG}} \propto E_{2\omega} \propto \chi^{(2)} E_\omega E_\omega + \chi^{(3)} E_\omega E_\omega \Phi(0) \qquad (1)$$

Early work, in which the relative phase of the terms contributing to the second harmonic generation (SHG) intensity was included[7], shows that when the wavelength of the fundamental and second harmonic photons are far from electronic and vibrational resonance, $\chi^{(2)}$ and $\chi^{(3)}$ are real, although they may differ in sign[7,8]. Yet, phase information has not been recovered in traditional SHG detection schemes, as they only collect the square modulus of the signal. Although phase information from SHG and vibrational sum frequency generation (SFG) signals can be readily obtained through coherent interference of the signal of interest with an external[9-16] or internal[17,18] phase standard, applications of such reference techniques to determine the phase of SHG signals generated at buried interfaces, such as charged oxide/water interfaces, is challenging due to the presence of dispersive media on both sides of the interface. In addition, the interface between water and α-quartz, the most abundant silicate mineral in nature[19-21], has been theoretically predicted to produce a more ordered interfacial water layer than amorphous silica[22-24], although this has not yet been probed using even traditionally detected SHG, Shen and co-workers applied nonlinear optics under resonant conditions to the α-quartz/water interface but did not report measurements taken at different rotational angles[18,23,25]. as the non-centrosymmetric bulk generally produces second harmonic signals that overpower surface SHG signals by orders of magnitude to the point where the surface signal is indistinguishable from the bulk response. Doing so under non-resonant conditions, however, would circumvent interaction terms between possibly potential-dependent $\chi^{(2)}$ and $\chi^{(3)}$ contributions to which resonantly enhanced SHG[26] or SFG[16] studies of charged interfaces may be sensitive.

Here we present an experimental geometry that produces considerable non-resonant SHG signal intensity from the z-cut α-quartz/water interface in the presence of bulk SHG signals from both the quartz and the electrical double layer under conditions of dynamically varying pH and ionic strength (Fig. 1). The approach, which uses an external reflection geometry, femtosecond laser pulses having just nanojoule pulse energies and a high repetition rate, enables us to experimentally identify a source of surface potential-induced bulk SHG from the electrical double layer. Further, it expands the scope of SHG spectroscopy to probe interfaces of non-centrosymmetric materials and establishes phase-referenced SHG spectroscopy to buried interfaces by using z-cut α-quartz as an internal phase standard.

## Results

**pH jumps over silica and quartz/water interfaces.** Using our previously described dual-pump flow system[27], we transition the pH of the aqueous phase between pH 3 and 11.5 so as to probe the interfacial potential dependence of the SHG responses. Near the point of zero charge, reported variously in the literature as pH 2.2 (ref. 28) and pH 2.6 (ref. 29) for α-quartz and pH 2.3 for fused silica[30], little SHG signal from the interface is expected, whereas the considerable negative interfacial potential at pH 11.5 should yield considerable SHG signal intensity[1,27]. Figure 2 shows the SHG versus time traces obtained from the fused silica/water interface using the flow cell shown in Fig. 1. Indeed, we observe the same increases (respectively decreases) in SHG signal intensity on increasing (respectively decreasing) the bulk solution pH that we previously reported for fused silica using a total internal reflection geometry (see Supplementary Fig. 1)[27]. At 3 mM total salt concentration, SHG signal intensities generally range between 5 and 10 counts per second at pH 3, and between 30 and 40 counts per second at pH 11.5. On replacing the fused silica window with right-handed z-cut α-quartz oriented 30° from the +x axis (see Methods), we observe the same response, albeit with a large bulk signal intensity leading to both significantly larger overall signal intensities and variations in signal intensity with varying pH.

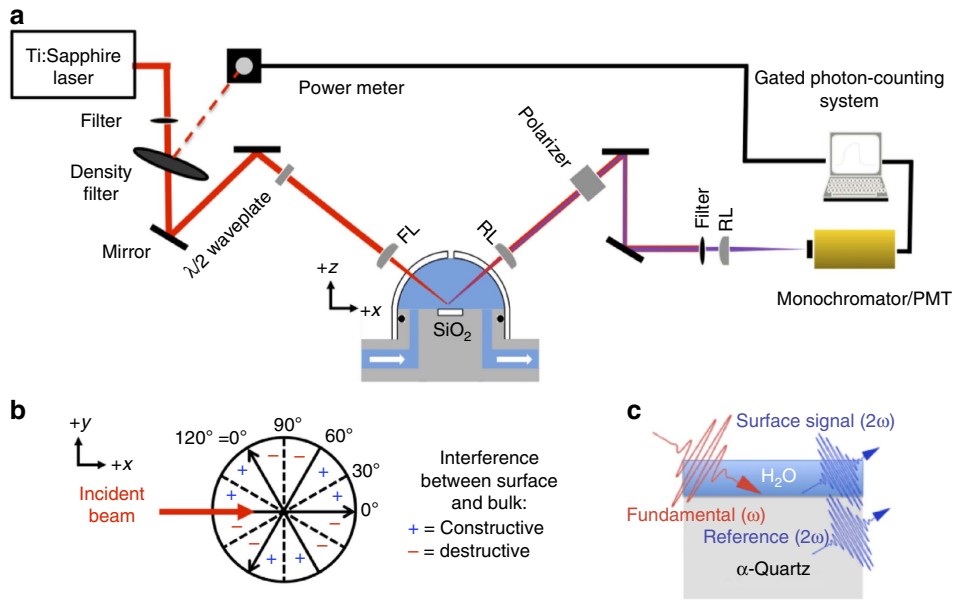

**Figure 1 | Experimental setup.** (**a**) Depiction of the external reflection geometry and the flow cell enabling this work. FL, focal lens; RL, recollimating lens. (**b**) Top view of right-handed z-cut α-quartz as placed across the plane of incidence and depiction of the sign of the interference between surface and bulk signals. (**c**) Depiction of the phase-referencing process used to unveil the newly identified term $i\chi^{(3)} \times \Phi(0)$.

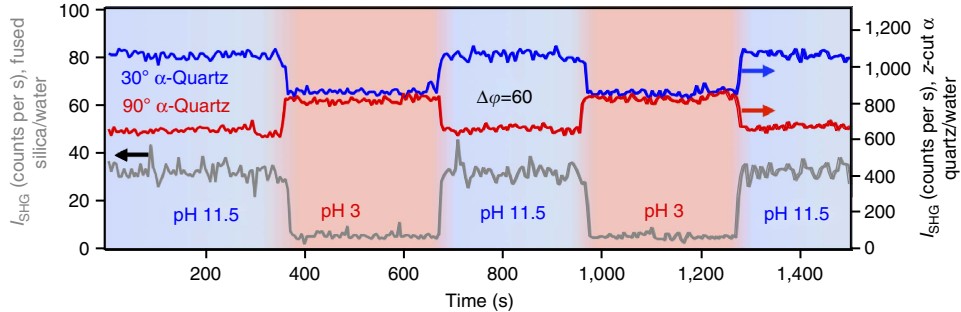

**Figure 2 | pH jump experiments.** SHG intensity in counts per second obtained from the fused silica/water (grey trace, left y axis) and α-quartz/water (blue and red traces, right y axis) interfaces under conditions of dynamically changing bulk solution pH varying between 3 and 11.5. Blue and red trace indicate results obtained with 60° difference in α-quartz rotational angle. The constructive and destructive interference observed for the α-quartz/water interface confirm interference between the potential-dependent interfacial and bulk quartz terms of the nonlinear susceptibility. Salt concentration = 3 mM = constant.

**Rotating quartz orientation angle reveals interference.** Figure 2 shows that the change in the SHG intensity observed for high versus low pH flips sign on rotation of the quartz crystal by 60° around the z axis, indicating modulation between negative and positive interference with the $\chi^{(2)}$ term of the bulk quartz. The P-in/P-out polarization combination (termed 'PP' hereafter) was selected, as it demonstrated the highest interfacial sensitivity out of the PP, PS, SP and 45S polarization combinations surveyed (see Supplementary Fig. 2). Figure 3 further shows that the constructive and destructive interference of the SHG signal depends on changing crystal rotation angles for the PP polarization combination; Supplementary Fig. 3 shows the $\Delta I_{SHG}$ as a function of quartz rotational angle. As in this geometry the beams must propagate through the dynamically changing aqueous phase, the dependence of the observed changes in the SHG intensity on the rotational angle of the α-quartz substrate indicates their origin as the interface and not changing optical properties of the aqueous phase, which would not depend on the angle of the α-quartz substrate. Additional control experiments show invariance of the results with minor variations in focal lens position (Supplementary Fig. 4) and quadratic dependence of the $I_{SHG}$ on laser power (Supplementary Fig. 5), as expected.

## Discussion

The constructive and destructive interference seen in Figs 2 and 3 can be understood by recalling that the non-resonant SHG or SFG signal from bulk α-quartz is a purely imaginary term[31,32]. This property of α-quartz has been employed to provide an internal phase standard that can amplify and interfere with the imaginary part of the vibrational SFG spectra of molecular surface species[17,18]. Moreover, vibrational SFG spectra of the α-quartz/water interface have been reported[18,23] but not for measurements taken at different rotational angles. Yet, when the surface second-order susceptibility is non-resonant, that is, all the surface response terms are purely real, interference with the imaginary term of the bulk α-quartz response cannot occur. However, the observations shown in Fig. 2 demonstrate that the non-resonant SHG signal from the water/α-quartz interface is subject to interference from the non-resonant SHG signal from the bulk α-quartz, indicating a new source of surface potential-induced bulk SHG from the aqueous solution.

The observed interference shown in Fig. 2 can be rationalized by considering that the phase of the bulk SHG signal produced by α-quartz shifts by 180° when the crystal is rotated clockwise by 60° around the z-axis. Rotation is shown to change the coherent interaction between the interfacial signal and bulk signal from constructive to destructive interference (the sign of the interference

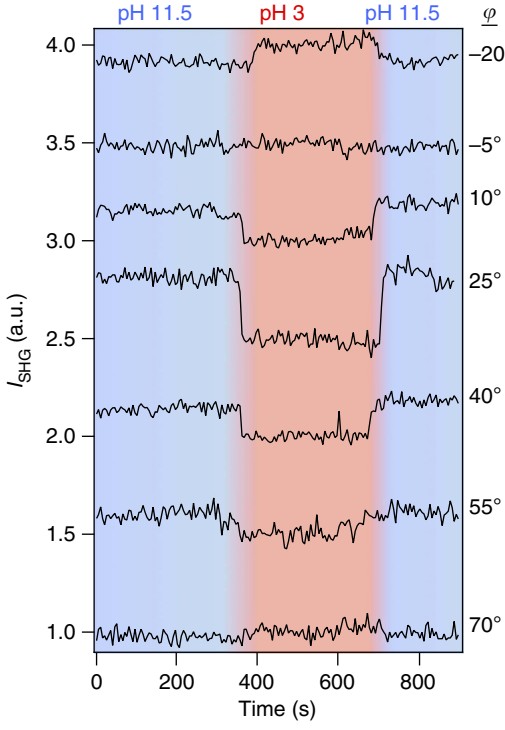

**Figure 3 | Interference experiment.** SHG intensity versus time traces normalized to intensity at pH 3 obtained from the α-quartz/water interface at different rotational angles of the α-quartz crystal during conditions of dynamically changing bulk solution pH varying between 3 and 11.5.

is depicted schematically in Fig. 1b for our geometry), as is expected from a 180° phase shift in the bulk quartz $\chi^{(2)}$ term[17]. This behaviour can be mathematically treated according to the following equation:

$$I_{SHG} \propto \left| \chi^{(2)} + \left( \chi_1^{(3)} - i\chi_2^{(3)} \right) \Phi(0) \pm i\chi_{\text{bulk quartz}}^{(2)} \right|^2 \quad (2)$$

where the sign of the $\pm i\chi_{\text{bulk quartz}}^{(2)}$ term is controlled by the rotational angle of the α-quartz crystal. Here, $\chi_1^{(3)}$ and $i\chi_2^{(3)}$ are related by a phase-matching factor as described in Supplementary Note 1. Even though the $\chi^{(3)}$ mechanism for interfacial potential-induced SHG has been long established[1], the importance of phase matching in the $\chi^{(3)}$ term has only recently been theoretically considered[33,34]. An experimental validation

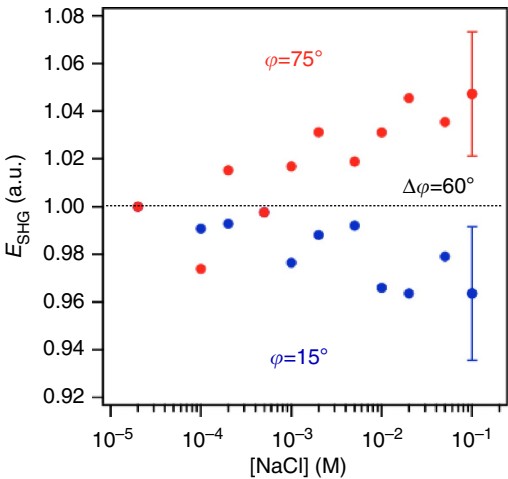

**Figure 4 | Salt screening experiment.** Normalized SHG intensity from the α-quartz/water interface maintained at pH 7 during conditions of dynamically changing bulk solution ionic strength varying between $10^{-5}$ and $10^{-1}$ M NaCl. Error bars represent 1 s.d., with six measurements.

requires a phase-referenced measurement, similar to the one demonstrated and established herein.

As further confirmation that the observed changes in the SHG signal intensity with pH are attributable to the interface, we recorded SHG signal intensities at pH 7 under conditions of increasing ionic strength for two quartz crystal rotation angles differing by 60°. For a charged interface, increases in the ionic strength result in screening of the interfacial charges, thus reducing the interfacial potential to which the water molecules in the electrical double layer are subjected and ultimately the associated SHG signal intensity[5,34]. Indeed, this behaviour is observed for both fused silica (Supplementary Fig. 6) and α-quartz (Fig. 4), at ionic strengths above $\sim 10^{-4}$ M NaCl, with opposite behaviour (increases in the ionic strength coincide with SHG intensity increases) observed on rotating the crystallographic axis of the z-cut quartz by 60°, as is observed in the pH jump experiments. We thus confirm the constructive and destructive interference discovered here for the α-quartz/water interface.

Our findings greatly expand the scope of SHG spectroscopy beyond amorphous and centrosymmetric materials, and towards crystal classes that lack centrosymmetry, including the >500 non-centrosymmetric oxides catalogued to date[35]. In doing so, they open a path for directly comparing the amphoteric properties of amorphous and crystalline materials, such as fused silica and α-quartz. The implications of using an SHG bulk signal as an internal reference from which phase information of the surface signal—and therefore orientation information at the interface—can be determined are intriguing. The use of thin film deposition techniques such as atomic layer deposition, electron beam deposition or spin coating, or surface functionalization methods such as silane chemistry, directly on α-quartz or other reference materials with known phase and second-order susceptibilities, offers the option of using phase-referenced SHG spectroscopy as a method for unambiguously determining the sign ( + or − ) of surface charges. A thin film with a thickness of only a few nanometres would be expected to interact with water molecules in a similar manner to the surface of its bulk material, yet still allow coherent interaction between the second harmonic signal generated at the interface and the bulk signal produced by the α-quartz substrate. Likewise, a thin film of a nonlinear optical crystal with a known phase grown on fused silica or another optically transparent centrosymmetric medium would allow for the

presence of an internal phase standard with an acceptable SHG intensity without requiring propagation through an aqueous medium. Yet, for conditions of electronic or vibrational resonance, we caution that the absorptive (imaginary) and dispersive (real) terms of $\chi^{(2)}$, $\chi_1^{(3)}$ and $\chi_2^{(3)}$ may mix.

## Methods

**Sample information.** In the experiments, we worked with three different right-handed, z-cut α-quartz samples (10 mm diameter and 3 mm thick) from three different vendors: Meller Optics (Providence, RI), Knight Optical (North Kingston, RI), and Precision Micro-Optics (Woburn, MA). The fused silica sample was purchased from Meller Optics. Before measurements, the samples were treated for 1 h with NoChromix solution (Godax Laboratories), a commercial glass cleaner (caution: NoChromix should only be used after having read and understood the relevant safety information). The samples were then sonicated in methanol for 6 min, sonicated in Millipore water for 6 min, dried in a 100 °C oven and plasma cleaned (Harrick Plasma) for 30 s on the highest setting. This procedure produces surfaces that are void of vibrational SFG responses in the C–H stretching region[27].

**Determining crystal orientation.** Owing to the dependence of the $I_{SHG}$ response on the orientational angle of the α-quartz crystal sample, it was necessary to unambiguously identify the crystal orientation used in the experiments. In this study, we define φ to be the clockwise rotation of the crystal about its z axis, measured from its +x axis (that is, at 0° the incoming laser beam is aligned with its horizontal projection along the +x axis of the α-quartz crystal, at 30° the crystal has been rotated 30° clockwise and so on). The x axis of the crystal can be determined by finding the $I_{SHG}$ maximum in the PP polarization combination or the $I_{SHG}$ minimum in the PS polarization combination[17] (Supplementary Fig. 7), while rotating the crystal about its axis. Determining the orientation of the x- axis, that is, whether the incoming laser beam is aligned parallel or anti-parallel with the x axis, is more difficult. Possible techniques include measuring the sign of the small voltage produced on deformation of the crystal due to its piezoelectricity[37], determining whether the bulk signal constructively or destructively interferes with the SFG C-H stretching signal from alkane chain monolayers absorbed on the interface[17] or obtaining Laue diffraction patterns from the α-quartz crystal. We compared Laue diffraction patterns from an α-quartz crystal of known orientation (provided by the supplier) with that of our unknown sample, to determine its absolute orientation (Supplementary Fig. 8). We also obtained two commercial α-quartz samples for which the suppliers (Knight Optical and PM Optics) had determined and provided us the absolute crystal orientation and obtained the same SHG responses across all three samples (Supplementary Fig. 3b).

**Laser setup.** We focused the p-polarized 800 nm output of a Ti:Sapphire oscillator (Mai Tai, Spectra-Physics, 120 fs pulses, λ = 800 nm, 82 MHz) through a hollow fused quartz dome onto the interface between water and the solid substrate in the external reflection geometry depicted in Fig. 1. Before the sample stage, the beam was passed through a long-pass filter to remove any second harmonic co-propagating with the fundamental beam, attenuated with a variable density filter to 0.50 ± 0.01 W, passed through a half-wave plate for input polarization control and focused onto the α-quartz/water or fused silica/water interface at an angle of 60°. The beam waist in the focal region is estimated at 30 μm in diameter.

The SHG signal was recollimated, passed through a 400 nm bandpass filter (FBH400-40, Thor Labs) and directed through a polarizer and monochromator for detection via a Hamamatsu photomultiplier tube connected to a preamplifier (SR445A, Stanford Research Systems) and a single photon counter (SR400, Stanford Research Systems), as detailed in our earlier work[27]. A portion of the fundamental beam was picked off before the sample stage and continuously monitored by a power meter (Newport 1917-R) during acquisitions, to allow for continuous normalization of signal intensity to power and account for the impact of slight, albeit unavoidable, laser power fluctuations on the SHG signal intensity. Compared with internal reflection, SHG signals obtained using the present geometry are generally $\sim 400$ times more sensitive to the interface relative to the bulk (see Supplementary Table 1).

We note that geometries in which 120 fs pulses from a kHz amplifier laser system accessed the α-quartz/water interface through bulk quartz as thin as 200 μm were not successful, even when applying a second quartz plate for background suppression[38].

**Solution preparation.** The aqueous solutions were prepared using Millipore water (18.2 MΩ cm$^{-1}$) and NaCl (Alfa Aesar, 99 + %). The pH of the solutions was adjusted using dilute solutions of HCl (E.M.D., ACS grade) and NaOH (Sigma Aldrich, 99.99%) and verified using a pH meter.

**Data availability.** All relevant data are available from the authors upon request to the corresponding author.

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

## Acknowledgements

Signal sensitivity analysis and measurement performed by P.E.O., S.A.S. and F.M.G. was supported by the U.S. National Science Foundation (NSF) under graduate fellowship research programme (GRFP) awards to P.E.O. and S.A.S., and award number CHE-1464916 to F.M.G. Method design/development by F.M.G. and S.A.S. was supported by the U.S. Department of Energy, Office of Science, Office of Basic Energy Sciences, Chemical Sciences, Geosciences, and Biosciences Division. H.F.W. was supported by the Materials Synthesis and Simulation Across Scales (MS3) Initiative through the LDRD programme at Pacific Northwest National Laboratory (PNNL). PNNL is a multi-programme national laboratory operated for Department of Energy by Battelle under Contract Number DE-AC05-76RL01830. K.B.E. gratefully acknowledges NSF award number CHE-1057483.

## Author contributions

F.M.G., H.-F.W. and K.B.E. conceived of the idea. P.E.O. and S.A.S. performed the experiments. F.M.G., H.-F.W. and P.E.O. analysed the data. The manuscript was written with substantial contributions from F.M.G., H.-F.W., P.E.O. and S.A.S.

## Additional information

**Competing financial interests:** The authors declare no competing financial interests.

**Publisher's note**: 

