## [Peer Review File · Nature Communications]

Reviewers' comments:

Reviewer #1 (Remarks to the Author):

This is an interesting, well-written paper describing original studies from top figures in the field of interface spectroscopy. It is likely to be an important and influential paper, and I can recommend that it be published in Nature Comm after the following issues have been addressed:

There are a few trivial typos in the abstract and text that should be fixed. Read carefully!

The authors are not very clear on how their crystal axes are oriented relative to the experimental setup. They discuss the 60 degree rotation, but fail to mention how they initially configured their experiment. In the end this may not be critical, as from figure 3, it looks as if maybe any rotation will achieve their results, but it would be helpful to add details in terms of reproducing their results.

Reference 22, which states that the components of the z cut quartz are always imaginary, seems to be wrong. The paper cited doesn't seem to mention quartz at all.

Given that the potential-dependent χ^3 term arises from the net orientation of the water near the surface, and given that the experiments are done off-resonance, it's interesting that they are measuring an imaginary component. They should explain this. See <http://pubs.acs.org/doi/abs/10.1021/acs.jpcc.5b12453>

Reviewer #2 (Remarks to the Author):

Summary:

The authors investigate experimentally the α -quartz/water interface using second harmonic generation (SHG) spectroscopy. This is an extension of the previous investigation (Achtyl et al. Nature Comm. 2015), where the use of a non-centrosymmetric material was introduced which enabled the control of the phase of the signal by rotating the crystal. The authors compare the results with those obtained with fused silica, which is used as a reference, as well as vary the ionic strength of the solution. The paper concludes that the observed signal originates from the water/quartz interface, as confirmed by the performed control experiments, and propose the possibility of implementing this method for studying the interface with non-centrosymmetric crystals by using the SHG bulk signal as an internal reference.

General Comments:

The manuscript is nicely written, concise, well-structured and contains high quality results, which are discussed to the point. The paper discusses the technical advancement of SHG spectroscopy by implementing a non-centrosymmetric α -quartz crystal, which enables precise control of the phase of the observed signal. This is a major improvement in the field of SHG spectroscopy, which opens up the possibility of studying a large variety of interfaces by using non-centrosymmetric crystals and non-resonant conditions. I find that the phase $\Delta\phi$ value discussed between figures 2 and 3 needs to be clarified in more detail and I encourage the author to reconsider their formulation. Also, it will help validate the experimental results and claims if the authors discussed the polarization and intensity dependence. Therefore, I invite the authors to address these major comments and technical concerns that will strengthen the manuscript before a final decision is reached.

Major Comments:

1. The authors mention that the phase shifts by 180 degrees when rotating the sample by 60 degrees along z. However, this is not clear in my view from figure 2, where is shown the signal for different $\Delta\phi$ angles. Is the $\Delta\phi$ in figure 2 the difference between two crystal angles or the absolute value? In the former case looks like 40 degrees is actually larger in amplitude than 60. In the latter case looks like the signal is actually changing phase every 90 degrees. This point can be better clarified by introducing an additional panel where is plotted the difference signal $\Delta I = I(\rho H11) - I(\rho H3)$ for different $\Delta\phi$ angles.

2. In the setup at figure 1 is shown a $\lambda/2$ waveplate and a polarizer along the beam path, which however is not referenced in the text. How does the signal phase depend on the polarization of the incident beam? One can imagine that in the case of quartz/water interface this can potentially be an additional way to control the phase of the observed signal.

3. An additional way to estimate the degree of mixing between χ_2 and χ_3 is to measure the signal for different laser intensities (flux dependence). Have the authors attempted such a measurement? In any case, the values that were used in the current realizations (both on the quartz and silica) and the focus size should be reported.

Minor comments:

1. From a technical point of view this is a very important breakthrough. It is also as important however to discuss what can be concluded about the water/quartz interface. Can one probe differences between the interaction of water with quartz versus that with fused silica? Varying the ionic strength is a good approach, which can potentially provide such an insight.

2. Similar investigations using sum frequency generation (SFG) have been performed to probe resonantly water/oil [Scatena, L. F.; Brown, M. G.; Richmond, G. L. *Water at Hydrophobic Surfaces: Weak Hydrogen Bonding and Strong Orientation Effects*. *Science* 2001, 292, 908–912.], water/lipid [Mondal, J. A.; Nihonyanagi, S.; Yamaguchi, S.; Tahara, T. *Structure and Orientation of Water at Charged Lipid Monolayer/ Water Interfaces Probed by Heterodyne-Detected Vibrational Sum Frequency Generation Spectroscopy*. *J. Am. Chem. Soc.* 2010, 132, 10656–10657] and water/solid interfaces [Tian, C. S.; Shen, Y. R. *Structure and Charging of Hydrophobic Material/Water Interfaces Studied by Phase-Sensitive Sum-Frequency Vibrational Spectroscopy*. *Proc. Natl. Acad. Sci. U. S. A.* 2009, 106, 15148–15153.]. Can the authors comment on the similarities and differences between the two approaches?

3. It would be helpful to show in figure 1 the axis of rotation z, by implementing a 3D representation like those previously (Achtly et al. *Nature Comm.* 2015).

4. The water thickness that was used in 3mm, whereas as a possible outlook is mentioned that this layer can be reduced to a few nm. What are the technical limitations for such a realization? This could help estimate the bulk contribution to the signal and additionally quantify the thickness sensitivity of SHG spectroscopy.

Reviewer #1 (Remarks to the Author)

Comments:

This is an interesting, well-written paper describing original studies from top figures in the field of interface spectroscopy. It is likely to be an important and influential paper, and I can recommend that it be published in Nature Comm after the following issues have been addressed:

We very much appreciate the reviewer's comments and are addressing the points raised as outlined below.

There are a few trivial typos in the abstract and text that should be fixed. Read carefully!

We apologize for the initial oversight and have carefully read and corrected typos in the abstract and main text.

The authors are not very clear on how their crystal axes are oriented relative to the experimental setup. They discuss the 60 degree rotation, but fail to mention how they initially configured their experiment. In the end this may not be critical, as from figure 3, it looks as if maybe any rotation will achieve their results, but it would helpful to add details in terms of reproducing their results.

We agree with the reviewer that clarifying the absolute orientation of the α -quartz crystal during experimentation will be beneficial to readers. We have replaced the $\Delta\phi$ formulation with the absolute orientation of the crystal, ϕ , the clockwise rotation of the crystal about its z-axis, measured from its +x-axis (*i.e.* at 0° the incoming laser beam is aligned with its horizontal projection along the +x-axis of the α -quartz crystal, at 30° the crystal has been rotated 30° clockwise, etc). We have added this information to the main text and changed the figures to include this absolute orientation, as well as added information on determining the absolute orientation of an α -quartz crystal to the SI: Methods Section SIII, Figure S7, and Figure S8. As the reviewer suggests may be the case, we believe that the absolute orientation of the crystal is less important than the periodicity and the fact that constructive and destructive interference is seen at all.

Reference 22, which states that the components of the z cut quartz are always imaginary, seems to be wrong. The paper cited doesn't seem to mention quartz at all.

We agree with the reviewer that this point could use further clarification. The paper in question (Byrnes, S. J.; Geissler, P. L.; Shen, Y. R. *Chem. Phys. Lett.* **2011**, 516 (4–6), 115-124) discusses the theoretical basis of surface and bulk contributions to nonlinear susceptibilities in general. In Eq. 1, describing the interaction between surface and bulk contributions, there is an additional factor of i associated with bulk term. This comes from Maxwell's equations and is also discussed elsewhere, for example (N. Bloembergen, P.S. Pershan, *Phys. Rev.*, 128, 606, 1962; and Kemnitz, K.; Bhattacharyya, K.; Hicks, J. M.; Pinto, G. R.; Eisenthal, B.; Heinz, T. F. *Chem. Phys. Lett.* **1986**, 131 (4–5), 285-290). As the input and output beams are far from resonance with the bulk material in our experimental setup, its bulk signal would be purely real, with this 90° phase shift making it instead, purely imaginary. We have added the Bloembergen reference to the main text as it is the original explicit treatment of the subject.

Given that the potential-dependent chi 3 term arises from the net orientation of the water near the surface, and given that the experiments are done are off-resonance, it's interesting that they are measuring an imaginary component. They should explain this. See <http://pubs.acs.org/doi/abs/10.1021/acs.jpcc.5b12453>

The imaginary component comes from the phase matching factor associated with the depth dependence of the static E-field at the interface; it is nicely derived in the paper mentioned and we have also included a derivation in SI Note 1. We have included a reference to the derivation by Gonella *et al.* in the main text. Despite the derivation of this phase matching factor, the presence and implications of an imaginary term are not explicitly mentioned by Gonella *et al.* As this imaginary component from the aqueous interfaces will also interfere with the reference signal produced in heterodyne-detected SFG experiments from buried interfaces, complicating their interpretation, the existence of this term, experimentally measured here for the first time, has broad implications and merits the explicit consideration given in this manuscript.

Reviewer #2 (Remarks to the Author):

Summary:

*The authors investigate experimentally the α -quartz/water interface using second harmonic generation (SHG) spectroscopy. This is an extension of the previous investigation (Achtyl *et al.* Nature Comm. 2015), where the use of a non-centrosymmetric materials in introduced which enable the control of the phase control of the signal by rotating the crystal. The authors compare the results with those obtained with fused silica, which is used as a reference, as well as vary the ionic strength of the solution. The paper concludes that the observed signal originates from the water/quartz interface, as confirmed by the performed control experiments, and propose the possibility of implementing this method for studying the interface with non-centrosymmetric crystals by using the SHG bulk signal as an internal reference.*

General Comments:

The manuscript is nicely written, concise, well-structured and contains high quality results, which are discussed to the point. The paper discusses the technical advancement of SHG spectroscopy by implementing a non-centrosymmetric α -quartz crystal, which enables to precise control of the phase of the observed signal. This is a major improvement in the field of SHG spectroscopy, which opens up the possibility of studying a large variety of interfaces by using non-centrosymmetric crystals and non-resonant condition. I find that the phase $\Delta\phi$ value discussed between the figure 2 and 3 needs to be clarified in more detail and I encourage the author to reconsider their formulation. Also, it will help validate the experimental results and claims if the authors discussed the polarization and intensity dependence. Therefore, I invite the authors to address these major comments and technical concerns that will strengthen the manuscript before a final decision is reached.

We very much appreciate the reviewer's comments and address the points regarding the $\Delta\phi$ value, the polarization, and intensity dependence in order below.

Major Comments:

 1. The authors mention that the phase shifts by 180 degrees when rotating the sample by 60 degrees along z. However, this is not clear in my view from figure 2, where is shown the signal for different $\Delta\phi$ angles. Is the $\Delta\phi$ in figure 2 the difference between two crystal angles or the absolute value? In the former case looks like 40 degrees is actually larger in amplitude than 60. In the latter case looks like the signal is actually changing phase every 90 degrees. This point can be better clarified by introducing an additional panel where is plotted the difference signal $\Delta I = I(\text{pH11}) - I(\text{pH3})$ for different $\Delta\phi$ angles.

We apologize for any confusion resulting from Figure 3, which shows the angle dependence of the interfacial response. We have followed the reviewer's recommendation and eliminated the $\Delta\phi$ formulation entirely. We have replaced the $\Delta\phi$ formulation with the absolute orientation of the crystal, ϕ , the clockwise rotation of the crystal about its z-axis, measured from its +x-axis (*i.e.* at 0° the incoming laser beam is aligned with its horizontal projection along the +x-axis of the α -quartz crystal, at 30° the crystal has been rotated 30° clockwise, etc). We have added this information to the main text and changed the figures to include this absolute orientation, as well as added information on determining the absolute orientation of an α -quartz crystal to the SI: Methods Section SIII, Figure S7, and Figure S8. We thank the reviewer for the suggestion of plotting ΔI as a function of rotational angle, which we have included in the SI (Figure S3, included below) and helps to demonstrate the flip from constructive to destructive interference that occurs with a 60° rotation for not just one, but three independently obtained samples.

The main text now refers to this plot in the top paragraph of page 5.

Figure S3. The difference in I_{SHG} at low and high pH conditions, ΔI_{SHG} , as a function of rotational angle of the α -quartz crystal for (A) only the data set included in Figure 3 in the main text, and (B) for all experiments carried out on three different samples from different suppliers (blue circles, PM Optics; open black circles, Meller Optics Experimental Run 1; closed black circles, Meller Optics Experimental Run 2; red circles, Knight Optical).

2. In the setup at figure 1 is shown a $\lambda/2$ waveplate and a polarizer along the beam path, which however is not referenced in the text. How does the signal phase depend on the polarization of the incident beam? One can imagine that in the case of quartz/water interface this can potentially be an additional way to control the phase of the observed signal.

The polarization states of the incoming and outgoing beams control which elements of the nonlinear susceptibility contribute to the measured response, and thus control the rotational dependence of the signal intensity from the bulk α -quartz crystal. We have added a figure to the SI (Figure S7, included below) showing this polarization dependence for the quartz/air interface.

Figure S7A: Rotational angle dependence of I_{SHG} for PP (blue, open triangles) and PS (red, upside down closed triangles) of bulk α -quartz measured in air using the reflection geometry described in the Supplementary Methods section.

The pH jump experiments were performed in the PP, SP, PS, and 45S polarization combinations; the phase was not found to change with different polarizations but the intensity of the jump was the highest in the PP polarization combination (Figure S1, below).

In addition to this figure, we have added the sentence in the main text: “The PP polarization combination was selected as it demonstrated the highest interfacial sensitivity out of the PP, SP, PS, and 45S polarization combinations (see SI Figure S2).” to the main text.

Figure S1: Polarization combination dependence of the α -quartz I_{SHG} response to pH jump experiments (the first index represents input polarization, the second represents output polarization; 45 represents mixed polarization).

3. An additional way to estimate the degree of mixing between χ_2 and χ_3 is to measure the signal for different laser intensities (flux dependence). Have the authors attempted such a measurement? In any case, the values that were used in the current realizations (both on the quartz and silica) and the focus size should be reported.

A systematic study of pH jump intensity versus input power has not been performed; however, the dependence of I_{SHG} on input power at constant pH was measured for the α -quartz/water interface to verify quadratic dependence (see below). For all other experiments, the power was set at 0.50 ± 0.01 W and continuously monitored as described in the SI Section SII. The beam waist in the focal region is estimated at $30 \mu\text{m}$. We have updated the Methods section in the Supplementary Information accordingly.

Figure S5: The power dependence of I_{SHG} for the α -quartz/water interface at pH 3; exponent in power fit = 2.1(1) as expected for a second order process such as SHG.

Minor comments:

 1. From a technical point of view this is a very important breakthrough. It is also as important however to discuss what can be concluded about the water/quartz interface. Can one probe differences between the interaction of water with quartz versus that with fused silica? Varying the ionic strength is a good approach, which can potentially provide such an insight.

We appreciate these comments and agree with the reviewer that beyond the implications to SHG spectroscopy, this technique allows important experimental access to the α -quartz/water interface. In particular, it has been theoretically predicted that the repeating crystalline structure of α -quartz could induce a more highly ordered interfacial water layer than that adjacent to an amorphous material such as fused silica (see for example Ostroverkhov, V.; Waychunas, G. A.; Shen, Y. R. *Chem. Phys. Lett.* **2004**, *386* (1–3), 144-148 and Cimas, Á.; Tielens, F.; Sulpizi, M.; Gaigeot, M.-P.; Costa, D. *J. Phys.: Condens. Matter* **2014**, *26* (24), 244106). We have added this description as a motivation behind studying this system in the main text.

We are thus excited to be able to probe both interfaces directly, and discern differences between them. The initial studies reported here show no difference in behavior between α -quartz and fused silica in terms of finding high SHG signal intensities at high pH and low intensities at low pH (provided the appropriate quartz crystal orientation). A systematic study comparing the behavior of the two substrates across a variety of pH and ionic strength conditions is in progress and will be reported in due course. We have added the specific phrase “In doing so, they open a path for directly comparing the amphoteric properties of amorphous and crystalline materials, such as fused silica and α -quartz.” to the future outlooks section.

2. Similar investigations using sum frequency generation (SFG) have been performed to probe resonantly water/oil [Scatena, L. F.; Brown, M. G.; Richmond, G. L. *Water at Hydrophobic*

Surfaces: Weak Hydrogen Bonding and Strong Orientation Effects. Science 2001, 292, 908–912.], water/lipid [Mondal, J. A.; Nihonyanagi, S.; Yamaguchi, S.; Tahara, T. Structure and Orientation of Water at Charged Lipid Monolayer/ Water Interfaces Probed by Heterodyne-Detected Vibrational Sum Frequency Generation Spectroscopy. J. Am. Chem. Soc. 2010, 132, 10656–10657] and water/solid interfaces [Tian, C. S.; Shen, Y. R. Structure and Charging of Hydrophobic Material/Water Interfaces Studied by Phase-Sensitive Sum-Frequency Vibrational Spectroscopy. Proc. Natl. Acad. Sci. U. S. A. 2009, 106, 15148–15153.]. Can the authors comment on the similarities and differences between the two approaches?

SFG is a powerful tool to probe interfacial vibrational stretching characteristics. Unlike the technique described here, it requires pico- to femtosecond pulses of both IR and VIS light, which must be overlapped in space and time at the interface of interest. When performing phase sensitive measurements, an additional reference SFG signal must also be overlapped, which can be technically very challenging at buried interfaces, and sensitive to phase drift. Our results indicate that SFG spectra of charged interfaces, like the systems the reviewer refers to, are subject to phase interference not only between the various oscillators, given by $\chi^{(2)}$, but also the interfacial potential, through $i\chi^{(3)}$.

The technique described here is much more straightforward to implement, as only one beam, from a stable and relatively cheap oscillator (Mai Tai, Spectra Physics) is required, and detection is done using a PMT. Most importantly, the spatial and temporal overlap of a reference signal is achieved without the use of additional optics and free from phase drift by the presence of a reference nonlinear optic (the bulk of the α -quartz crystal) directly adjacent to the interface of interest. Though off-resonant SHG spectroscopy does not provide the chemical bond specificity of vibrational SFG spectroscopy, it has still been demonstrated to be a powerful tool to probe adsorption/desorption at interfaces, charge state and interfacial potentials, and, with a phase reference, molecular orientation.

3. It would be helpful to show in figure 1 the axis of rotation z, by implementing a 3D representation like those previously (Achtly et al. Nature Comm. 2015).

We have made the schematic of the sample cell to more accurately represent the actual cell, as well as added a schematic of the crystal and appropriate axis designations for each. Though it remains in 2D, we hope it gives a clearer sense of the experimental setup. If it remains unclear, we can continue to try to improve this graphic.

4. The water thickness that was used in 3mm, whereas as a possible outlook is mentioned that this layer can be reduced to a few nm. What are the technical limitations for such a realization? This could help estimate the bulk contribution to the signal and additionally quantify the thickness sensitivity of SHG spectroscopy.

The nanometer thickness mentioned in the future outlook section referred to the thickness of a thin film on the surface of the α -quartz, like one that could be prepared using atomic layer deposition, physical vapor deposition, surface functionalization through silanization, or spin coating, not the thickness of the water layer.

The hollow hemispherical dome has an interior diameter of 2.54 cm, so the incident beam passes through approximately 1.27 cm before reaching the sample. The electrical field produced by the interface interacts with water molecules up to a length characterized by the Debye length (see Gonella, G., Lütgebaucks, C., de Beer, A. G. F. & Roke, S. *J. Phys. Chem. C* **120**, 9165-9173, **2016**), and this whole region contributes to the $\chi^{(3)}$ term of the surface susceptibility. This is typically on the scale of nanometers, though at very low concentrations, this can extend up to the micron length scale. Though doing a study with varying thicknesses of water in order to experimentally determine how many water molecules contribute to the nonlinear response is intriguing, controlling the depth of water present to that level or precision would be challenging, and would likely be influenced by the nature of the second interface (air, silica, etc.), the effects of which would be difficult to separate out.

REVIEWERS' COMMENTS:

Reviewer #1 (Remarks to the Author):

The concerns of the referees have been addressed in impressive detail, and the paper is now suitable for publication.

Reviewer #2 (Remarks to the Author):

In my view the authors have addressed the referees comments in great detail and the manuscript has been improved significantly. This is a fine piece of work and will make a great publication at nature communications.